# Correlation of the antibacterial activity of commercial manuka and *Leptospermum* honeys from Australia and New Zealand with methylglyoxal content and other physicochemical characteristics

Kathryn J. Green[1,2], Ivan L. Lawag[2,3], Cornelia Locher[2,3], Katherine A. Hammer[1,2,4]*

**1** School of Biomedical Sciences, The University of Western Australia (UWA), Crawley, Western Australia, Australia, **2** Cooperative Research Centre for Honey Bee Products Limited (CRC HBP), Yanchep, Western Australia, Australia, **3** Division of Pharmacy, School of Allied Health, UWA, Crawley, WA, Australia, **4** The Marshall Centre for Infectious Diseases Research and Training, School of Biomedical Sciences, UWA, Crawley, WA, Australia

* Katherine.hammer@uwa.edu.au

**Data Availability Statement:** All relevant data are within the paper and its Supporting Information files.

## Abstract

Variation in the antibacterial potency of manuka honey has been reported in several published studies. However, many of these studies examine only a few honey samples, or test activity against only a few bacterial isolates. To address this deficit, a collection of 29 manuka/*Leptospermum* honeys was obtained, comprising commercial manuka honeys from Australia and New Zealand and several Western Australian *Leptospermum* honeys obtained directly from beekeepers. The antibacterial activity of honeys was quantified using several methods, including the broth microdilution method to determine minimum inhibitory concentrations (MICs) against four species of test bacteria, the phenol equivalence method, determination of antibacterial activity values from optical density, and time kill assays. Several physicochemical parameters or components were also quantified, including methylglyoxal (MGO), dihydroxyacetone (DHA), hydroxymethylfurfural (HMF) and total phenolics content as well as pH, colour and refractive index. Total antioxidant activity was also determined using the DPPH* (2,2-diphenyl-1-picrylhydrazyl) and FRAP (ferric reducing–antioxidant power) assays. Levels of MGO quantified in each honey were compared to the levels stated on the product labels, which revealed mostly minor differences. Antibacterial activity studies showed that MICs varied between different honey samples and between bacterial species. Correlation of the MGO content of honey with antibacterial activity showed differing relationships for each test organism, with *Pseudomonas aeruginosa* showing no relationship, *Staphylococcus aureus* showing a moderate relationship and both *Enterococcus faecalis* and *Escherichia coli* showing strong positive correlations. The association between MGO content and antibacterial activity was further investigated by adding known concentrations of MGO to a multifloral honey and quantifying activity, and by also conducting checkerboard assays. These investigations showed that interactions were largely additive in nature, and that synergistic interactions between MGO and the honey matrix did not occur.

**Funding:** This research was funded by the Cooperative Research Centre for Honey Bee Products (Project 12 - KH and Project 13 - CL). https://www.crchoneybeeproducts.com/ The funders had no role in study design, data collection and analysis, decision to publish, or preparation of the manuscript.

**Competing interests:** The authors have declared that no competing interests exist.

## Introduction

Manuka honey, which is defined as honey derived from the nectar of *Leptospermum scoparium* (J.R. Forster & G. Forster) flowers, is well known for its antibacterial activity. The activity is termed "non-peroxide" as hydrogen peroxide is not a significant antimicrobial factor in these honeys, in contrast to many other types of honey. Following the identification of methylglyoxal (MGO) as a major antibacterial compound within manuka honey [1,2], several studies have quantified MGO content in manuka honeys and have shown that it correlates strongly with antibacterial activity determined by the agar diffusion "phenol equivalence" assay [1,3,4]. The phenol equivalence assay was developed in the early 1990s in New Zealand, and has since been used in both commercial and research laboratories to quantify the antibacterial activity of honey. Measurements obtained using this method include Total Activity (TA), which includes antibacterial activity due to hydrogen peroxide, and non-peroxide activity (NPA), which quantifies the antibacterial activity remaining when any hydrogen peroxide is removed and is also referred to as the Unique Manuka Factor (UMF). Manuka honey, and to a lesser extent MGO, has been shown to exert several antibacterial actions, including inducing changes in cell morphology and preventing cell division [5,6]. Both manuka honey and MGO ultimately cause the death of microorganisms when their concentrations exceed tolerable levels [7]. In addition to MGO, physicochemical characteristics (e.g. pH and osmotic activity) or other components (e.g. phenolic compounds and proteins) within manuka honeys contribute to its antibacterial action [8,9].

Whilst considerable scientific data have been published on manuka honeys, those studies describing its antibacterial activity have several limitations. For example, studies analysing large numbers of manuka honeys have utilised the phenol equivalence assay to quantify activity, which uses a one specific strain of the Gram positive *Staphylococcus aureus*, meaning that results may not be broadly generalisable [1,3,4]. Also, the assay relies on the diffusion of antibacterial compounds through agar, which is known to be problematic for many natural products that may not be water soluble [10]. Lastly, the assay has limited sensitivity [4] and cannot detect activity in all honeys. On the other hand, those studies providing an in-depth analysis of manuka honey's antibacterial activity, often using the broth dilution assay or time kill assays, have typically only investigated one or two manuka honey samples, and may not include MGO data to correlate with the antibacterial data, which limits the interpretation of results. For example, Girma *et al*. (2019) [11] examined the antibacterial activity of three commercial manuka honeys that were sold with antibacterial activity levels of UMF 5+, 10+ and 15+ using a broth microdilution assay, and found that the antibacterial activity was highest in the honey with the lowest UMF rating, directly contradicting the UMF levels provided on the product labels [11]. However, MGO levels on the honeys were not quantified, so activity could not be correlated with these. The aim of this study was therefore to investigate a relatively large collection of manuka/*Leptospermum* honey samples against several different bacterial species using a number of susceptibility testing techniques, and to correlate these data with MGO content and other physicochemical parameters.

## Materials and methods

### Honey samples

A total of 30 honeys were examined, including 25 commercial manuka honeys from Australia and New Zealand, four Western Australian (WA) *Leptospermum* honeys and a commercial multifloral honey (Capilano Honey Ltd, Western Australia), with no floral source specified. All manuka honeys and the multifloral honey were purchased from retailers, whereas the WA

**Table 1. Floral source, country of origin and MGO levels as stated on product labels, for manuka/*Leptospermum* honeys and comparators.**

| Study Code | Honey Type | MGO content[a] | Country of Origin[b] | Floral Source |
|---|---|---|---|---|
| MN01 | Manuka | 800 | NZ (South Island) | *L. scoparium* |
| MN02 | Manuka | not stated | AUS | *Leptospermum* sp.* |
| MN03 | Manuka | 900 | AUS (Eastern states) | *Leptospermum* sp. |
| MN04 | Manuka | 120 | AUS (Eastern states) | *L. polygalifolium* (Salisbury) |
| MN05 | Manuka | 400 | AUS | Not provided |
| MN06 | Manuka | 263 | AUS | *L. scoparium* |
| MN07 | Manuka | 514 | AUS | *L. scoparium* |
| MN08 | Manuka | 830 | AUS | *L. scoparium* |
| MN09 | Manuka | 30 | AUS | Not provided |
| MN10 | Manuka | 800 | NZ (Northern) | *L. scoparium* * |
| MN11 | Manuka | 400 | NZ | *L. scoparium* * |
| MN12 | Manuka | 550 | NZ | *L. scoparium* * |
| MN13 | Manuka | 250 | AUS (Tasmania) | *L. scoparium* |
| MN14 | Manuka | 83 | AUS | Not provided |
| MN15 | Manuka | 100 | AUS | Not provided |
| MN16 | Manuka | 400 | NZ | *Leptospermum* sp.* |
| MN17 | Manuka | 30 | AUS | *Leptospermum* sp. |
| MN18 | Manuka | 75 | AUS | *Leptospermum* sp. |
| MN19 | Manuka | 30 | AUS (Southwest WA) | *L. scoparium* |
| MN20 | Manuka | 125+ | AUS (Southwest WA) | *L. scoparium* |
| MN21 | Manuka | 300+ | AUS | Not provided |
| MN22 | Manuka | 550+ | AUS | Not provided |
| MN23 | Manuka | 250+ | AUS | Not provided |
| MN24 | Manuka (NPA 5+) | 83+ | AUS | Not provided |
| MN25 | Manuka | - | NZ | *L. scoparium* * |
| MN26 | *Leptospermum* | - | AUS (WA) | *Leptospermum* sp. (endemic) |
| MN27 | *Leptospermum* | - | AUS (WA) | *Leptospermum* sp. (endemic) |
| MN28 | *Leptospermum* | - | AUS (WA) | *Leptospermum* sp. (endemic) |
| MN29 | *Leptospermum* | - | AUS (Southwest WA) | *Leptospermum* sp. (endemic) |
| MUL | Multifloral | - | Western Australia | not provided |
| ART | Artificial | - | not applicable | not applicable |

[a] MGO units are mg/kg.

[b] New Zealand (NZ); Australia (AUS); Western Australia (WA).

* The floral source is implied rather than explicitly stated (e.g. "New Zealand manuka honey").

*Leptospermum* honeys were obtained directly from beekeepers. Information including country of origin, relevant flowering species, MGO content (mg/kg) and measures of antibacterial activity such as "Non-Peroxide Activity" and/or "Unique Manuka Factor" was obtained from jar labels, or directly from the beekeeper (Table 1). The exceptions were honeys MN10 and MN25, for which data was obtained from the company websites. Of the 25 manuka honeys, six were from New Zealand and the remaining 19 were from different locations within Australia. Honeys were all well within the expiry or 'best before' dates stated on the labels. All honeys were stored in their original packaging in the dark at room temperature ($22 \pm 1°C$) for the duration of the study, and were analysed within three months of acquisition. Artificial honey was prepared as described previously by Cooper et al. [12] and was stored alongside the other honeys.

## Quantitation of physicochemical parameters

The methylglyoxal (MGO), dihydroxyacetone (DHA) and hydroxymethylfurfural (HMF) content of all manuka and *Leptospermum* honeys was determined using a previously published method [13] and the corresponding (theoretical) non-peroxide activity (NPA) was then calculated from the quantified level of MGO. Temperature-adjusted Refractive Index and Brix values were determined simultaneously by spreading a sample of each honey over the entire surface of the reading window of a digital refractometer (Hanna Instruments, Smithfield, RI, USA) as per the instrument manual. Honeys that had crystallised or contained small bubbles were subsampled into glass bottles and gently heated by placing in a water bath at 50°C for no more than 4 h until completely homogeneous. Samples were then cooled to room temperature ($22 \pm 1$°C) before determining values. Some honeys did not completely dissolve, or small bubbles did not dissipate after heat treatment meaning that these values could not be determined. The pH of each honey was measured by dissolving 1 g of honey in 7.5 ml of carbon-dioxide free water [14] then determining the pH with a calibrated pH meter (A211 Benchtop pH Meter, Orion Star). To quantify colour, solutions of 50% (w/v) honey were prepared in sterile distilled water and then the optical density (OD) was measured at 450 nm and 720 nm [14] using a spectrophotometer (SpectraMax 190, Molecular Devices, San Jose, California, USA). The difference between the OD measurements was multiplied by 1000 and expressed as milli-absorbance units (mAU). Colour was determined for all honeys both before and after passing through a 0.7 μm glass fibre filter, which was used to remove debris or particles that could potentially interfere with the OD readings.

**Hydrogen peroxide generation.** Hydrogen peroxide levels were determined using o-dianisidine and horseradish peroxidase reagents as described elsewhere [8]. Briefly, each honey was dissolved in sterile distilled water at a final concentration of 30% (w/v) [15]. Honey solutions were held at room temperature ($22 \pm 1$°C), and after 1, 2, 4, 6 and 24 h aliquots were removed and o-dianisidine and horseradish peroxidase reagents were added. The reaction was stopped after 5 min by the addition of 6 M sulfuric acid and the OD was determined at 540 nm. Blanks for each honey contained all reagents except the o-dianisidine and horseradish peroxidase. A hydrogen peroxide standard curve was generated in each experiment using doubling dilutions of hydrogen peroxide solution ranging from 550–2.1 μM, with additional standards containing 440 and 330 μM $H_2O_2$ included to improve the accuracy and linearity of the standard curve [16]. The level of $H_2O_2$ in each honey was then determined from the $H_2O_2$ standard curve.

**Total phenolics content.** The total phenolics content of triplicate samples of honey was determined as described previously [17,18]. In brief, a standard curve was prepared by spiking artificial honey (prepared according to Bobis et al., [19] with gallic acid standards. Aqueous honey solutions were reacted with Folin-Ciocalteu reagent at slightly alkaline pH to supress interference from reducing sugars. Absorbance was determined at 760 nm after 2 h using artificial honey solution to blank the instrument (UV-Vis-Cary 50 Bio UV-Visible Spectrophotometer, Agilent Technologies, Santa Clara, CA, USA). Using the gallic acid standard curve, total phenolic content was expressed as gallic acid equivalent (GAE) per 100 g of honey.

**Ferric Reducing Antioxidant Power (FRAP) assay.** The FRAP assay was carried out as described previously [17]. In brief, aqueous honey solutions (20% w/v) were reacted in triplicate with FRAP reagent and the antioxidant activity determined at 620 nm using a POLARstar Optima (BMG Labtech, Allmendgrün, Ortenberg, Germany) Microplate Reader. The antioxidant activity was expressed as mmol $Fe^{2+}$/ kg of honey against a standard curve of $FeSO_4 \cdot 7H_2O$ that ranged in concentration from 1200 μM to 200 μM.

**2,2-Diphenyl-1-picryl-hydrazyl-hydrate (DPPH) free radical assay.** The scavenging ability of 2,2-diphenyl-1-picrylhydrazyl (DPPH) radicals was also used to determine the antioxidant activity of honeys. As described previously [17], aqueous solutions of honey (20% w/v) were reacted in triplicate with DPPH* reagent at pH 5.5. After being kept in the dark for 2 h, the absorbance was measured at 520 nm using a POLARstar Optima (BMG Labtech, Allmendgrün, Ortenberg, Germany) microplate reader. Antioxidant activity, derived from a standard curve of Trolox solutions ranging in concentration from 100 to μM, was expressed as μmol Trolox equivalent per kg of honey.

## Determination of antibacterial activity

Minimum inhibitory concentrations (MICs) of each honey were determined using two Gram positive and two Gram negative quality control reference strains recommended by the Clinical and Laboratory Standards Institute (*Staphylococcus aureus* ATCC 29213, *Escherichia coli* ATCC 25922, *Enterococcus faecalis* ATCC 29212 and *Pseudomonas aeruginosa* ATCC 27853) using a broth microdilution method [20] as described previously [21]. Briefly, a 40% (w/v) honey solution was prepared in distilled water, filter sterilised, then aliquoted in appropriate volumes into wells of a 96 well microtitre plate. After the addition of 50 μl of inoculum to each well, final honey concentrations ranged from 2 to 30%, in 2% increments in final well volumes of 200 μl. A positive growth control containing no honey was included. Inocula were prepared in quadruple strength Mueller Hinton Broth (4 × MHB) to account for the dilution factor when adding the inoculum to the honey solutions in each well. After incubation, MICs were determined visually as the lowest concentration of honey preventing visible growth. In addition, the optical density of each tray well was determined at 600nm before and after incubation. Initial ODs were subtracted from 24 h ODs, then all OD values were expressed as a percentage of the positive growth control.

MICs of an MGO solution (Sigma M0252) were also determined using the broth microdilution method [20] and the organisms mentioned above. MGO was prepared such that after inoculation, final concentrations ranged in doubling dilutions from 4.096 mg/mL to 0.004 mg/mL. A positive growth control without MGO was included. To quantify the effect of incremental increases in MGO content on the antibacterial activity of honey, MICs of multifloral honey with additional MGO were determined using the broth microdilution method described above. To produce a honey with the desired MGO content (mg/kg), honey was weighed out and the appropriate volume of MGO solution (10 mg/mL in sterile distilled water) was then added. The volume of MGO solution required was calculated from the exact weight of the honey and the desired final concentration of MGO. The remaining volume was then made up with sterile distilled water to result in a 40% (w/v) honey solution amended with MGO, which was then dissolved and filter sterilised before use in the broth microdilution assay.

The antibacterial activity of all honeys was also quantified using a spectrophotometric broth assay to generate antibacterial activity values (AAVs) [21]. Briefly, four test bacteria were incubated in the presence of 30, 25, 20, 15, 10, 5 and 0% honey (w/v) in MHB, and OD values were determined at 600nm before and after incubation. The OD for each honey concentration was expressed relative to the positive growth control OD, and the previously described formula was applied to calculate the AAV. Antibacterial activity was also determined using the agar diffusion "phenol equivalence" assay as described previously [21,22]. The limit of detection in this assay, based on a theoretical inhibition zone size of 9 mm, was 7% phenol (which is equivalent to a TA of 7). Honeys with no detectable zone were given a value of <7 TA. To further examine the relationship between MGO concentration and zone size, solutions of MGO ranging from 0.001% - 10% MGO (corresponding to 10 mg/kg - 100g/kg) were tested in this same

assay by adding 100 μl volumes of each solution to wells and measuring zone diameters in mm after incubation.

All antibacterial activity assays were repeated at least twice on separate days. For MICs the mode was selected as the final value. Where there was no mode, the test was repeated, and the mode was selected or in the absence of a mode the arithmetic mean of replicate values was determined. For AAVs and phenol equivalence values the mean of replicate values was determined.

**Time-kill studies.** Three honeys with relatively high (1022 mg/kg), moderate (326 mg/kg) and low (75 mg/kg) MGO content were selected for examination in a time kill assay, and the multifloral honey with negligible MGO, and artificial honey with no MGO were tested in parallel for comparison. To prepare inocula, *S. aureus* ATCC 29213 and *E. coli* ATCC 25922 were cultured overnight at $36 \pm 1°C$ on blood agar, then 2–3 colonies were inoculated into a ~ 10 ml trypticase soy broth. These cultures were incubated for approximately 2 h at 37°C with shaking at 150 rpm to generate exponential phase cultures. Cultures were then adjusted to 3 McFarland in 0.85% saline, corresponding to approximately $9 \times 10^8$ cfu/ml. Erlenmeyer flasks (50 ml) were prepared containing appropriate volumes of $4 \times$ MHB, 60% (w/v) filter-sterilised honey solution and sterile distilled water such that after inoculation the final concentration of honey was 40% for *S. aureus* and was 30% for *E. coli*, and MHB was diluted to single-strength. At time zero, which was immediately after inoculation, a 100 μl aliquot was removed from the positive control flask for both organisms for viable counting. Flasks were incubated at 37°C with shaking at 120 rpm, and further samples were removed from all flasks after 2, 4 and 6 h for viable counts. Viable counting was performed by diluting samples 10-fold in 0.85% saline, then pipetting 20 μl volumes from each serial dilution dropwise in duplicate onto Mueller Hinton agar. After drops had absorbed, plates were incubated overnight at $36 \pm 1°C$. Colonies were counted and cell density in CFU/ml was calculated. The limit of detection was $2.5 \times 10^3$, based on the detection of five colonies in a 20 μl aliquot from the $10^{-1}$ serial dilution. The entire assay was repeated three times on separate days and final results were expressed as the mean $\log_{10}$cfu/ml.

**Determination of fractional inhibitory concentrations.** Preliminary data indicated that the concentrations of MGO present at the MIC of each manuka honey against each organism were well below the MIC of MGO alone, for each respective organism. For example, for a theoretical honey with MGO content of 500 mg/kg and an MIC of 10% honey, the MGO present at the 10% concentration would be only 50mg/kg, which is below the MIC of MGO alone. Given that the MGO may therefore be too dilute at the MIC of honey to have a direct antibacterial effect, it was thought that synergistic interactions could be occurring between MGO and the remaining honey matrix. To investigate this possibility, checkerboard assays were conducted using the two organisms *S. aureus* ATCC 29213 and *E. coli* ATCC 25922. The checkerboard assay was performed in 96-well microtitre trays using the broth microdilution methodology described above, with minor modifications. Dilutions of multifloral honey were prepared in 2% increments from 6% to 26%, with MGO added to each honey concentration in doubling dilutions from 0.004 to 0.256 mg/ml. A dilution series containing each agent (honey or MGO) alone was included, as was a positive growth control containing growth medium alone without any antimicrobial agent. After inoculation and incubation, fractional inhibitory concentrations (FICs) were calculated as described previously [23] and FICs were interpreted as synergistic, indifferent, additive or antagonistic. Assays were repeated at least twice on separate days.

## Statistical analysis

Unless stated otherwise, all physicochemical tests were repeated at least twice on separate days and the mean of replicate values was determined. For total phenolic and antioxidant assays, tests were repeated once with triplicate samples, from which mean values were determined. For antibacterial activity data, and for the purpose of analysis only, off-scale results were assigned specific values: MICs of >30% were assigned values of 32% and any TA value of <7% was assigned a value of 0%. To investigate relationships between antibacterial activity (MICs, TA and AAV) and physicochemical factors (including MGO), data was statistically analysed by determining Pearson correlation coefficients and generating a correlation matrix. Time kill data for *S. aureus* were analysed by repeated measures two-way ANOVA with Geisser-Greenhouse correction, followed by Tukey's multiple comparisons test. For *E. coli* time kill data, repeated measures ANOVA could not be performed due to missing values (lack of viable count data for MN03 at 4 and 6h). Therefore, *E. coli* data were analysed by Ordinary two-way ANOVA followed by Tukey's multiple comparisons test. Due to relatively low numbers of samples within specific subgroups of honeys, such as honeys from different countries (e.g. New Zealand compared to Australia) or from different floral sources (e.g. *L. scoparium* versus non-*scoparium*), these particular statistical comparisons could not be performed. All analyses were performed using GraphPad Prism (version 9.3.1).

## Results

### Physicochemical parameters

Levels of MGO quantified for manuka/*Leptospermum* honeys ranged from 3 mg/kg to 1022 mg/kg, with a median of 274 mg/kg (Table 2). For 16 (67%) of the 24 honeys with an MGO level stated on the label, the quantified MGO level varied by less than 100 mg/kg from that stated on the label. For several honeys, the quantified amount varied substantially from the stated MGO, from 503 mg/kg below (MN01) to 399 mg/kg above (MN23). DHA levels ranged from 21 to 1185 mg/kg, and HMF levels ranged from 15 to 432 mg/kg. Pearson correlation showed a strong relationship between MGO and DHA (r = 0.83; p < 0.05) and no relationship between HMF and either MGO or DHA (Table 3).

The pH of all manuka/*Leptospermum* honeys ranged from 3.93 to 5.42, with a median of 4.21 and mean of 4.28 (Table 2). The pH of multifloral honey was 4.24 and for artificial honey was 5.73. Brix values ranged from 79.8 to 84.9 (Table 2. Neither pH nor Brix showed strong relationships with any other physicochemical characteristics (Table 3). Colour after filtration for all manuka/ *Leptospermum* honeys ranged from 169 to 1351 mAU, with a mean of 580 mAU. Values for artificial and multifloral honeys were 44 and 209 mAU, respectively (Table 2). Colour showed a strong relationship with both HMF (r = 0.75) and total phenolics content (r = 0.79) (Table 3). The highest amount of hydrogen peroxide generated was 143 μM for honey MN28 (WA *Leptospermum*). All remaining hydrogen peroxide values ranged from 0–78 μM. Total phenolic content (GA eq.mg /100g honey) for manuka/*Leptospermum* honeys ranged from 21 to 66 with a mean of 35. Artificial honey and multifloral honey contained 1 and 24 GA eq.mg /100g, respectively.

Antioxidant activity measured by FRAP ranged from 2.9 to 10.7 mmol $Fe^{2+}$/kg for all manuka/*Leptospermum* honeys, whereas values for artificial honey and multifloral honeys were 0.75 and 4.0 mmol $Fe^{2+}$/kg, respectively. Antioxidant values quantified using DPPH* reagent, and expressed in μmol TE/kg at 2 h, ranged from 560 to 4345 for manuka/*Leptospermum* honeys, with a mean of 1933. For multifloral honey the value was 1559 and for artificial honey was <10. Antioxidant data obtained by FRAP and DPPH* assay showed strong correlation with

**Table 2. Physicochemical properties, MGO, DHA and HMF content, and antioxidant activity of manuka/*Leptospermum* and comparator honeys.**

| Honey Code | MGO (mg/kg) | DHA (mg/kg) | HMF (mg/kg) | pH | Refractive Index[a] | Brix[a] | $H_2O_2$ maximum (μM) | Colour (mAU) Before filtration | Colour (mAU) After filtration | Total Phenolics (GAE mg/kg) | Anti-oxidant activity FRAP (mmol Fe/kg) | Anti-oxidant activity DPPH (μmol TE/kg at 2h) |
|---|---|---|---|---|---|---|---|---|---|---|---|---|
| MN01 | 297 | 272 | 15 | 4.54 | 1.493 | 80.7 | 11 | 522 | 301 | 270 | 4.99 | 1825 |
| MN02 | 3 | 84 | 189 | 4.27 | 1.495 | 81.8 | 4 | 672 | 403 | 278 | 4.74 | 1816 |
| MN03 | 1022 | 630 | 112 | 4.08 | 1.493 | 80.9 | 5 | 1424 | 1036 | 431 | 6.64 | 2022 |
| MN04 | 174 | 157 | 57 | 4.31 | 1.500 | 83.4 | 11 | 465 | 287 | 253 | 4.29 | 1463 |
| MN05 | 326 | 243 | 120 | 4.21 | 1.492 | 80.4 | 4 | 1015 | 735 | 359 | 6.14 | 2353 |
| MN06 | 183 | 270 | 183 | 4.12 | 1.495 | 81.7 | 4 | 1782 | 1170 | 491 | 7.95 | 2974 |
| MN07 | 531 | 423 | 183 | 3.99 | 1.496 | 82.0 | 0 | 1537 | 1099 | 432 | 6.85 | 2568 |
| MN08 | 532 | 431 | 432 | 3.93 | B | B | 2 | 1701 | 1351 | 501 | 8.61 | 3230 |
| MN09 | 34 | 44 | 31 | 4.59 | 1.496 | 81.9 | 20 | 438 | 258 | 226 | 3.60 | 1224 |
| MN10 | 911 | 395 | 72 | 4.12 | B | B | 4 | 1162 | 613 | 425 | 7.22 | 2643 |
| MN11 | 494 | 482 | 36 | 4.20 | 1.491 | 80.3 | 2 | 686 | 391 | 353 | 6.56 | 2300 |
| MN12 | 572 | 448 | 34 | 4.17 | B | B | 2 | 709 | 395 | 340 | 6.30 | 2237 |
| MN13 | 280 | 259 | 57 | 4.23 | B | B | 1 | 1058 | 633 | 426 | 7.01 | 2760 |
| MN14 | 75 | 99 | 231 | 4.26 | 1.493 | 80.8 | 2 | 1434 | 1026 | 420 | 6.68 | 2318 |
| MN15 | 94 | 147 | 57 | 4.52 | 1.494 | 81.2 | 36 | 598 | 395 | 260 | 4.09 | 909 |
| MN16 | 575 | 348 | 68 | 4.11 | B | B | 9 | 1087 | 558 | 418 | 7.12 | 2465 |
| MN17 | 42 | 80 | 146 | 4.18 | C | C | 4 | 898 | 562 | 294 | 4.23 | 1158 |
| MN18 | 89 | 106 | 58 | 4.40 | 1.494 | 81.2 | 13 | 556 | 295 | 297 | 4.94 | 1776 |
| MN19 | 30 | 42 | 16 | 4.72 | 1.499 | 83.0 | 78 | 477 | 305 | 233 | 3.00 | 807 |
| MN20 | 124 | 163 | 78 | 4.20 | 1.492 | 80.7 | 5 | 866 | 507 | 356 | 5.64 | 1903 |
| MN21 | 454 | 610 | 82 | 4.14 | 1.491 | 80.1 | 4 | 681 | 451 | 258 | 2.88 | 560 |
| MN22 | 857 | 1185 | 46 | 4.15 | 1.490 | 79.8 | 5 | 661 | 418 | 292 | 3.56 | 1770 |
| MN23 | 649 | 898 | 23 | 4.27 | B | B | 4 | 478 | 271 | 231 | 3.02 | 1114 |
| MN24 | 111 | 206 | 173 | 4.30 | 1.498 | 82.7 | 0 | 650 | 373 | 263 | 3.46 | 1044 |
| MN25 | 274 | 196 | 109 | 4.00 | B | B | 0 | 1604 | 818 | 663 | 10.72 | 4345 |
| MN26 | 184 | 195 | 52 | 4.21 | 1.496 | 82.1 | 1 | 623 | 492 | 405 | 5.00 | 1394 |
| MN27 | 520 | 324 | 61 | 4.36 | 1.502 | 84.4 | 0 | 792 | 756 | 437 | 3.77 | 1294 |
| MN28 | 27 | 25 | 37 | 5.42 | 1.503 | 84.9 | 143 | 303 | 169 | 211 | 3.51 | 1099 |
| MN29 | 4 | 21 | 98 | 4.28 | 1.500 | 83.6 | 2 | 1168 | 749 | 444 | 5.86 | 2682 |
| MUL | not done | not done | not done | 4.24 | 1.491 | 82.4 | 0 | 431 | 209 | 244 | 3.99 | 1560 |
| ART | not done | not done | not done | 5.73 | 1.497 | 80.3 | 0 | 33 | 44 | 9 | 0.75 | <10 |

[a] B indicates an excess of bubbles; C indicates the honey remained crystallised after heating, both preventing an accurate reading.

each other (r = 0.96, p<0.05). FRAP and DPPH data also correlated strongly with total phenolics content, with r values of 0.89 and 0.87, respectively (Table 3) and also with colour (r values of 0.70 and 0.65 for FRAP and DPPH*, respectively).

## Antibacterial activity

A range of MICs was observed for the 29 manuka and *Leptospermum* honeys against the four test organisms, from relatively low (4%) to relatively high (30%). MICs for *S. aureus* ranged from 4% to >30% with a median of 8% (Table 4). For 17 of the 29 honeys (59%), MICs were ≤10%. MICs for *E. coli* ranged from 6% to 30% with a median of 16%. The median MIC for *P.*

**Table 3. Pearson correlation matrix showing relationships between physicochemical properties, antibacterial and antioxidant activities of manuka/*Leptospermum* honeys.**

| | 1 | 2 | 3 | 4 | 5 | 6 | 7 | 8 | 9 | 10 | 11 | 12 | 13 | 14 | 15 |
|---|---|---|---|---|---|---|---|---|---|---|---|---|---|---|---|
| **1. MGO** | - | | | | | | | | | | | | | | |
| **2. DHA** | 0.83 | - | | | | | | | | | | | | | |
| **3. HMF** | -0.02 | -0.07 | - | | | | | | | | | | | | |
| **4. pH** | **-0.47** | **-0.41** | **-0.43** | - | | | | | | | | | | | |
| **5. Brix** | -0.39 | -0.53 | -0.1 | **0.55** | - | | | | | | | | | | |
| **6. Colour After Filtering** | 0.27 | 0.08 | **0.75** | **-0.59** | -0.12 | - | | | | | | | | | |
| **7. Total Phenolics** | 0.26 | 0.01 | **0.43** | **-0.61** | -0.05 | **0.79** | - | | | | | | | | |
| **8. *S. aureus* MIC** | **-0.54** | **-0.4** | 0.34 | -0.03 | -0.18 | 0.02 | -0.14 | - | | | | | | | |
| **9. *E. coli* MIC** | **-0.87** | **-0.69** | 0.13 | **0.43** | 0.26 | -0.17 | -0.33 | **0.73** | - | | | | | | |
| **10. *E. faecalis* MIC** | **-0.94** | **-0.79** | -0.04 | **0.46** | 0.22 | -0.31 | -0.36 | **0.6** | **0.89** | - | | | | | |
| **11. *P. aeruginosa* MIC** | -0.08 | 0.05 | 0.19 | -0.28 | **-0.47** | 0.14 | -0.01 | **0.63** | 0.24 | 0.12 | - | | | | |
| **12. AAV** | **0.77** | **0.61** | -0.21 | -0.16 | -0.01 | 0.13 | 0.22 | **-0.83** | **-0.82** | **-0.8** | **-0.55** | - | | | |
| **13. Total Activity** | **0.55** | **0.39** | -0.24 | 0.14 | 0.33 | -0.11 | -0.11 | **-0.76** | **-0.54** | **-0.55** | **-0.63** | **0.72** | - | | |
| **14. FRAP** | 0.23 | -0.05 | **0.45** | **-0.55** | -0.29 | **0.7** | **0.89** | -0.11 | -0.33 | -0.29 | -0.05 | 0.17 | -0.07 | - | |
| **15. DPPH** | 0.22 | 0.01 | **0.41** | **-0.53** | -0.22 | **0.65** | **0.87** | -0.18 | -0.35 | -0.33 | -0.12 | 0.21 | -0.03 | **0.96** | - |

Shading indicates R values of ≥0.75 or ≤-0.75. Bold denotes statistical significance (P<0.05).

*aeruginosa* was 20% and MICs ranged from 10% to 26%, which was the smallest MIC range of the four test organisms. Similar to *S. aureus*, *P. aeruginosa* MICs did not show strong relationships with any physicochemical parameters. MICs for *E. faecalis* ranged from 12% to >30%, and the median was 26%, which was the highest of all the four organisms. MICs for the multifloral and artificial honeys ranged from 25 to >30% (Table 3). MICs for *E. coli* showed a relatively strong inverse relationship with both MGO (r = -0.87) and DHA (r = -0.69), as did MICs for *E. faecalis*, with r values of -0.94 with MGO and -0.79 with DHA. *S. aureus* MICs showed a weak inverse relationship with MGO (r = -0.54) and DHA (r = -0.40) whereas *P. aeruginosa* MICs showed no relationship with either (Table 3).

Heat maps of relative optical density data obtained from MIC assays for four selected manuka honeys are shown in Fig 1. These honeys represent different levels of MGO, with concentrations of 1022, 531, 280 and 124 mg/kg, for samples MN03, MN07, MN13 and MN20, respectively. Multifloral honey was also included as a comparison. The percentage OD relative to the positive growth control was calculated for each concentration of honey tested in the MIC assay. The heat maps show that honeys with higher concentrations of MGO exert greater inhibition of bacterial growth for *S. aureus*, *E. coli* and *E. faecalis* but not for *P. aeruginosa*. All manuka honeys inhibited growth to a greater extent than multifloral honey, for all organisms. Concentrations of honey resulting in 50% and 90% decreases in optical density relative to the positive growth control are shown in **S1 Table.**

The AAVs for all 29 manuka/*Leptospermum* honeys ranged from 189 to 651, with a median of 457 and a mean of 462 (Table 4). AAVs for multifloral and artificial honey were 148 and 194, respectively. AAVs showed a strong relationship with MGO and DHA, and also with MICs for *S. aureus*, *E. coli* and *E. faecalis*. The relationship with P. aeruginosa MICs was slightly weaker (r = -0.55).

Phenol equivalence or "Total Activity" values for the 29 manuka and *Leptospermum* honeys ranged from <9 to 30, with a median of 14 and a mean of 12. TA values of <7 were obtained for 12 of the 29 manuka/*Leptospermum* honeys (38%), as well as for the multifloral and artificial honeys. TA values showed a moderate relationship with MGO and DHA, but no other

**Table 4. Antibacterial activity of manuka/*Leptospermum* honeys and comparators, including minimum inhibitory concentrations, total activity and non-peroxide activity.**

| Honey Code | MIC (% w/v honey) | | | | Antibacterial Activity Value | Total Activity (% phenol) | Theoretical NPA (% phenol) |
|---|---|---|---|---|---|---|---|
| | *S. aureus* ATCC 29213 | *E. faecalis* ATCC 29212 | *E. coli* ATCC 25922 | *P. aeruginosa* ATCC 27853 | | | |
| MN01 | 12 | 26 | 15 | 21 | 460 | <7 | 10.8 |
| MN02 | >30 | >30 | 30 | 26 | 189 | <7 | 0.6 |
| MN03 | 6 | 12 | 8 | 16 | 647 | 22 | 22.7 |
| MN04 | 8 | 30 | 14 | 16 | 458 | 18 | 7.8 |
| MN05 | 12 | 22 | 17 | 26 | 446 | <7 | 11.4 |
| MN06 | 16 | 27 | 24 | 26 | 349 | <7 | 8.0 |
| MN07 | 8 | 19 | 12 | 22 | 517 | 16 | 15.3 |
| MN08 | 8 | 17 | 10 | 16 | 566 | 17 | 15.3 |
| MN09 | 9 | 32 | 26 | 16 | 412 | 16 | 2.9 |
| MN10 | 4 | 14 | 6 | 18 | 651 | 31 | 21.2 |
| MN11 | 8 | 19 | 10 | 20 | 537 | 20 | 14.6 |
| MN12 | 6 | 17 | 8 | 16 | 592 | 21 | 16.0 |
| MN13 | 12 | 24 | 16 | 26 | 425 | 10 | 10.4 |
| MN14 | 27 | >30 | 30 | 26 | 280 | <7 | 4.7 |
| MN15 | 27 | >30 | 28 | 26 | 413 | <7 | 5.4 |
| MN16 | 6 | 17 | 10 | 20 | 432 | 24 | 16.0 |
| MN17 | 29 | >30 | 29 | 21 | 314 | <7 | 3.3 |
| MN18 | 11 | >30 | 19 | 18 | 408 | <7 | 5.2 |
| MN19 | 6 | >30 | 28 | 15 | 441 | 22 | 2.7 |
| MN20 | 13 | >30 | 19 | 19 | 403 | <7 | 6.4 |
| MN21 | 8 | 21 | 13 | 26 | 493 | 13 | 13.9 |
| MN22 | 6 | 14 | 9 | 20 | 604 | 20 | 20.4 |
| MN23 | 6 | 14 | 10 | 18 | 581 | 21 | 17.3 |
| MN24 | 23 | >30 | 28 | 26 | 320 | <7 | 6.0 |
| MN25 | 9 | 20 | 12 | 16 | 518 | <7 | 10.3 |
| MN26 | 15 | 30 | 20 | 24 | 372 | <7 | 8.1 |
| MN27 | 8 | 16 | 11 | 22 | 539 | 11 | 15.1 |
| MN28 | 5 | 29 | 23 | 12 | 482 | 26 | 2.6 |
| MN29 | 6 | 27 | 24 | 10 | 554 | 19 | 0.7 |
| MUL | >30 | >30 | 29 | 25 | 148 | <7 | not done |
| ART | >30 | >30 | >30 | 28 | 194 | <7 | not done |

physicochemical characteristics. TA values showed a relatively strong relationship with both *S. aureus* MICs and AAV (Table 3). Zones sizes for MGO solutions were 55, 51, 46, 39 and 18 mm for MGO solutions of 10, 5, 2.5, 1 and 0.1%, respectively. No zones were produced by solutions of 0.01 or 0.001% MGO. Plotting the mean squared zone size against MGO concentration showed a strong linear relationship ($r^2 = 0.996$).

**Activity of MGO alone and combined with honey.** MICs of MGO alone were 128 mg/l for both *S. aureus* and *E. coli*, 256 mg/l for *E. faecalis*, and 512 mg/l for *P. aeruginosa* (Table 5). The addition of MGO to multifloral honey resulted in stepwise decreases in the MIC of honey for each organism as the concentration of MGO increased (Table 5). Responses varied between organisms, with *S. aureus* being the most sensitive to changes in MGO level and *P. aeruginosa* the least affected, with the MIC changing from 25% w/v honey without MGO, to 15% w/v honey at the highest MGO concentration of 1000 mg/kg. Changes in antibacterial activity with

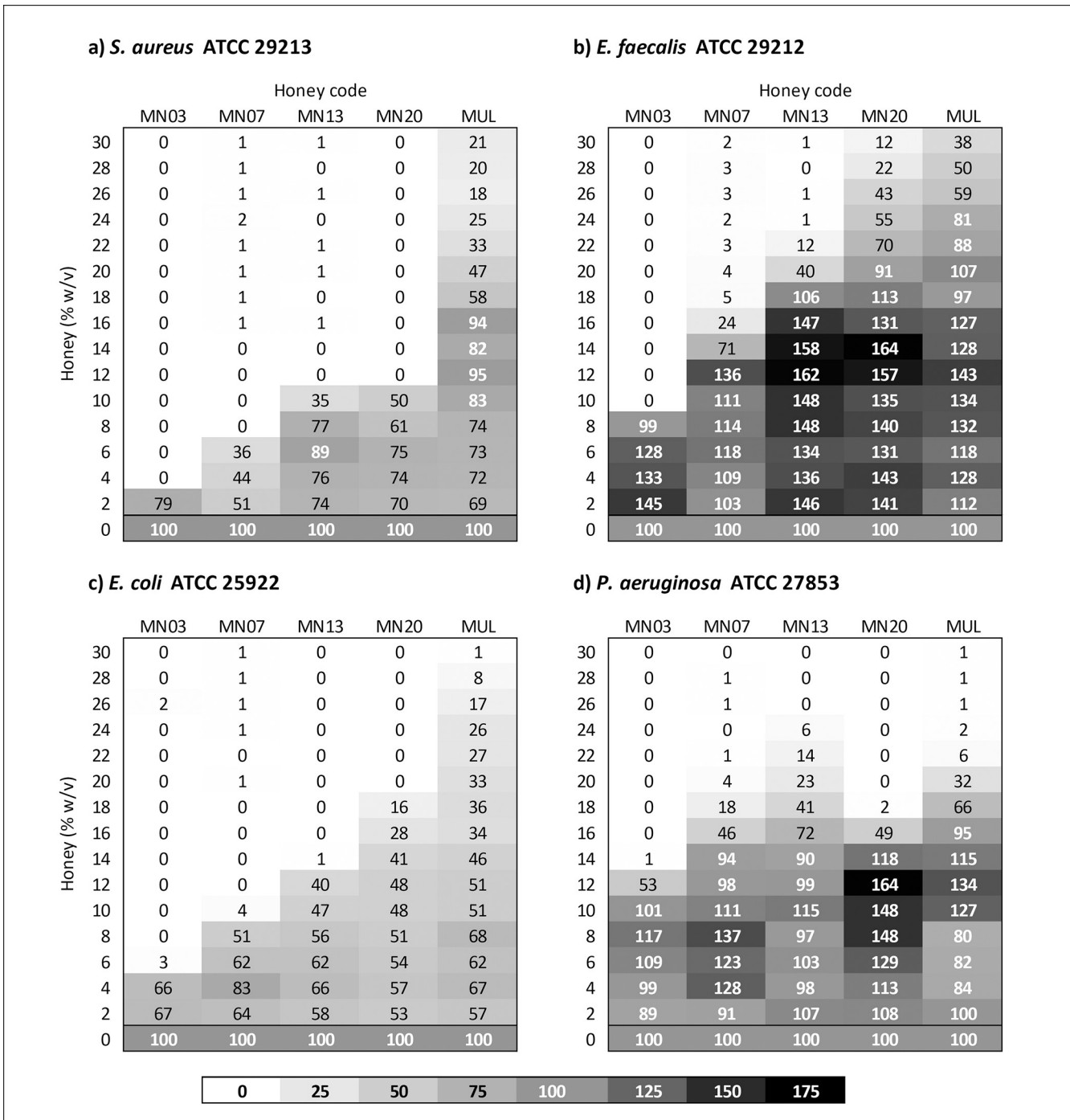

**Fig 1. Heat maps of relative optical density at 24 h for selected manuka honeys and multifloral honey.** Values indicate the relative optical density of wells containing honey compared to the positive control, expressed as a percentage.

increasing MGO concentration were also quantified by determining the AAV, whereby the AAV was 270 for honey with 50 mg/kg MGO and reached 647 at 1000 mg/kg MGO. Comparison of the antibacterial activity of each MGO-amended honey to a natural manuka honey containing similar MGO levels showed that antibacterial activity was approximately equivalent.

**Table 5. Antibacterial activity of multifloral honey amended with concentrations of MGO ranging from 50 to 1000 mg/kg.**

| Combination | Minimum inhibitory concentrations[1] | | | | AAV |
| --- | --- | --- | --- | --- | --- |
| | *S. aureus* ATCC 29213 | *E. faecalis* ATCC 29212 | *E. coli* ATCC 25922 | *P. aeruginosa* ATCC 27853 | |
| **MGO alone** | 128 mg/l | 256 mg/l | 128 mg/l | 512 mg/l | NA |
| **Multifloral honey alone** | >30% | >30% | 29% | 25% | 148 |
| **Multifloral + 50 mg/kg MGO** | 28% | >30% | 29% | 24% | 270 |
| **Multifloral + 100 mg/kg MGO** | 19% | >30% | 25% | 24% | 332 |
| **Multifloral + 250 mg/kg MGO** | 12% | 25% | 16% | 23% | 462 |
| **Multifloral + 500 mg/kg MGO** | 8% | 17% | 10% | 21% | 554 |
| **Multifloral + 750 mg/kg MGO** | 6% | 12% | 9% | 18% | 618 |
| **Multifloral + 1000 mg/kg MGO** | 4% | 10% | 6% | 15% | 647 |

[1] The units for MICs are mg/l for MGO and % w/v for honey alone and for the honey/MGO combinations.

Combination of multifloral honey with MGO in the checkerboard assay showed additive activity for *S. aureus*, with an FIC range of 0.69–1.19 and a median FIC of 0.91. Similarly, *E. coli* showed an FIC range of 0.75–1.25 and a median of 1.0, values also considered additive to indifferent. However, as the MIC of honey alone (without MGO) exceeded the highest test concentration, FICs were calculated from imputed rather than quantified values, which may have led to an inaccurate representation of activity. Heat maps of relative optical densities obtained from checkerboard assays are shown in Fig 2 and show the lack of synergistic action between MGO and multifloral honey.

**Time kill assays.** Viable counts for *S. aureus* treated with 40% honey (including artificial honey) differed significantly from the untreated control at each time point (Fig 3). In addition, at 4 h, the viable count for *S. aureus* treated with MN03 differed significantly from the multifloral honey (p = 0.041). At 6 h, viable counts for all honeys (including multifloral) differed

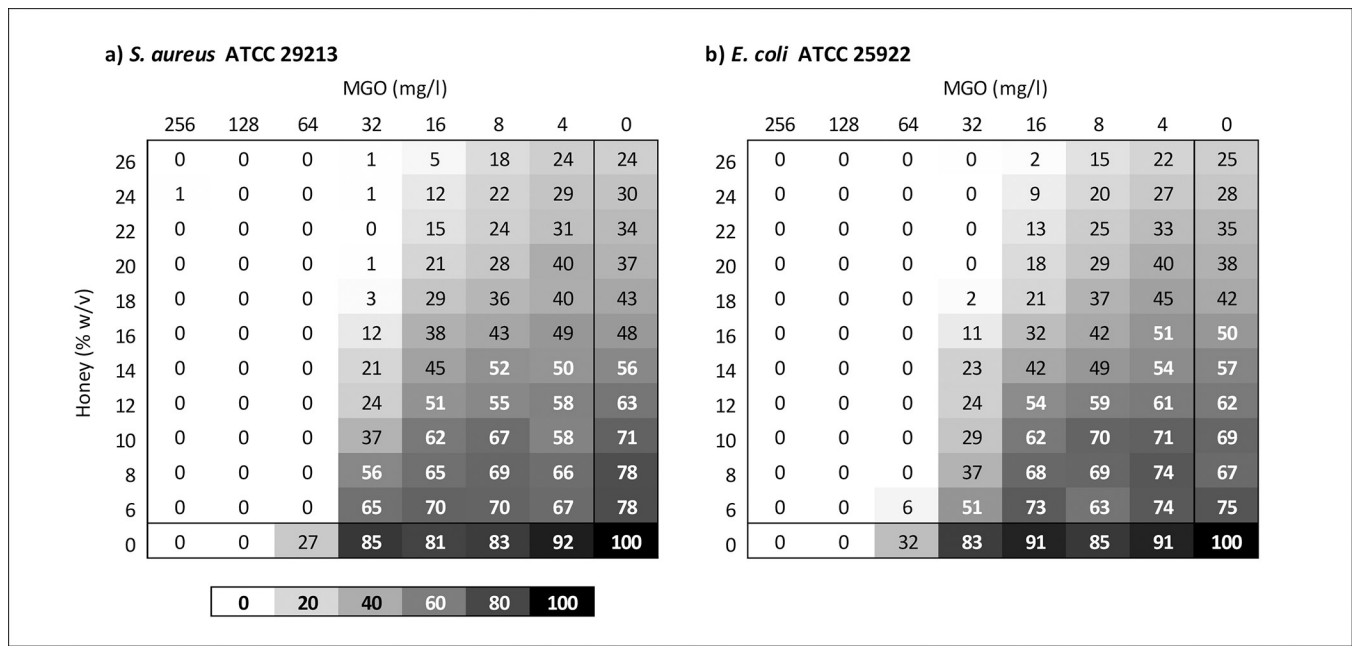

**Fig 2. Heat maps of checkerboards showing multifloral honey in combination with MGO.**

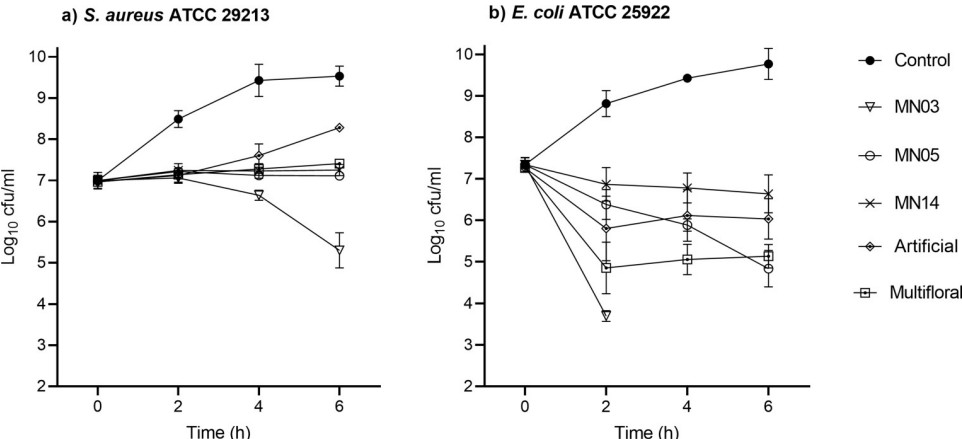

**Fig 3. Time kill curves of thee manuka honeys with varying MGO content, a multifloral honey and artificial honey against *S. aureus* ATCC 29213 and *E. coli* ATCC 25922.** MGO content was 1022, 326 and 75 mg/kg for honeys MN03, MN05 and MN14, respectively. For both organisms, viable counts for all honey treatments differed significantly from the untreated controls at each time point.

significantly from artificial honey, and in addition the viable count for MN03-treated *S. aureus* differed significantly from MN14, but not MN05 (p = 0.058). For *E. coli* treated with 30% honey, viable counts after treatment with all honeys (including artificial), differed significantly from the untreated control at each time point. Viable counts for individual honeys did not differ significantly from each other at any time point. Viable *E. coli* cells were below the limit of detection after treatment with honey MN03 from 4 h onwards.

## Discussion

This paper has investigated two questions that are pivotal to our understanding of the antibacterial activity of manuka honey. The first is how closely does antibacterial activity correlate with MGO content and the second is does MGO have an indifferent, additive or synergistic interaction with the remaining honey matrix with regard to antibacterial activity.

In order to address these questions, the honeys must also be characterised for physicochemical characteristics and phenolics content. Many different methods have been explored as tools to aid in the characterisation and authentication of different types of honey, including manuka [24]. Authentication may be required for detecting sugar syrup adulteration, or for authenticating the floral source, in which case the non-sugar fraction, including phenolic compounds, is typically assessed. One approach specifically applied to the authentication of manuka honey is to quantify levels of key compounds or biomarkers. Important compounds identified for manuka honey include 3-phenyllactic acid, 2'-methoxyacetophenone, 2-methoxybenzoic acid, 4-hydroxyphenyllactic acid, dihydroxyacetone (DHA), methylglyoxal (MGO), leptosperin and lepteridine, in addition to DNA from manuka pollen [25,26]. The compounds quantified in the current study, MGO and DHA, whilst not unique to manuka honeys, are present in substantially higher quantities than in non-manuka honeys. Previous studies show that MGO content can vary considerably between individual manuka honey samples, and may be as high as 800 mg/kg [2,3,27,28], as was also found in the current study. Of note in this study was that comparison of the MGO levels stated on the product label to the levels quantified showed a number of discrepancies, the majority of which were relatively minor. In several instances the level quantified was higher than the level stated on the label, which is likely due to the non-enzymatic conversion of DHA to MGO during honey storage [29]. In many of these particular

honeys, levels of DHA were higher than levels of MGO, indicating that the conversion process was indeed ongoing [27]. When the opposite occurred, and the quantified MGO level was substantially lower than that stated on the label, this could be due to loss or degradation of MGO due to heating at temperatures of 50°C or more [27,29,30]. Alternative explanations for these apparent losses of MGO in honey were not found in the scientific literature and it would seem that comprehensive investigations of factors affecting the long-term stability of MGO in honey have not been conducted. Interestingly, levels of HMF in most honeys were in excess of the maximum of 40 mg/kg (or 80 mg/kg for honeys from tropical climates) set by the Codex Alimentarius Standard commission. HMF is formed naturally in honey over time, and formation is accelerated by heating [31]. The levels of HMF may therefore indicate that the honeys have been stored for considerable time before being sold, or may have been heated during, or after processing. The honey with the highest HMF content (432 mg/kg) was also one of the honeys demonstrating a decrease in MGO content when comparing measured versus stated levels, which supports the theory that the honey may have been stored for considerable time. Since few studies have analysed large numbers of manuka/*Leptospermum* honeys for physicochemical properties such as pH, sugar content, colour, total phenolics content and antioxidant activity, comparison of the current data to previous studies is limited, however, the available data are broadly similar [9,32,33]. Manuka honeys produced very low levels of hydrogen peroxide, which was expected due to the known interference of MGO with the glucose oxidase enzyme [34].

Antibacterial activity experiments, including phenol equivalence and MIC assays, showed a range of activity across all honeys. Activity correlated with the MGO content of honeys to varying degrees, depending on the test organism and the assay. For the phenol equivalence assay, previous studies with manuka/*Leptospermum* honeys found a strong correlation between non-peroxide activity (NPA) and MGO content [1,3,4], whereas the correlation for honeys in the current study was moderate. This may be due to the relatively low number of samples tested here compared to these previous studies, which all tested more than 50 honeys each [1,3,4,22]. It may also be due to greater heterogeneity of samples in the current study, as they were sourced from multiple countries and *Leptospermum* species, and may well have contained multiple nectar sources. For these latter honeys, these may contain minor antibacterial components other than MGO, meaning that the zones of inhibition in the assay resulted from both MGO and other unquantified, antibacterial factors, which has also been suggested previously by others [1,4]. An example of newly discovered factor that may influence antibacterial activity is RNA, with recent studies showing that a range of small RNA fragments can be found in honey. This RNA, which may include small RNAs derived from invertebrates or prokaryotes [35], or plant-derived microRNA [36] has been shown to be intact and theoretically functional, thereby having a range of potential actions. The attribution of activity to unidentified antibacterial factors is further supported by the apparent mismatch between measured TA and theoretical NPA for some honeys. Theoretical NPA values, which were calculated based on MGO content alone, were in some instances substantially lower than the measured TA values, suggesting that these honeys likely contain additional antibacterial compounds.

Honey activity was also evaluated by generating AAVs from optical density data. Similar to the phenol equivalence assay, this assay also generates a single value to represent antibacterial activity but in contrast, utilises a broth medium instead of agar and utilises four test bacteria instead of one. Whilst there are few published data obtained using this method, results obtained here for manuka honeys were similar to those published for the honeys Jarrah (*Eucalyptus marginata* [Smith]) and Marri (*Corymbia calophylla* [(Lindl.) Hill & Johnson)] [21], which are regarded as having high antibacterial activity. In agreement with the study by Green

et al (2020) [22], AAVs in this study correlated moderately with the MIC of each organism and correlated well with measured MGO.

Further investigation of antibacterial activity using a broth microdilution assay showed that MICs obtained for honeys varied from relatively high to relatively low, and were generally comparable to previously published data [37–39]. Similar to these previous studies, the most sensitive test bacterium was *S. aureus*, which had both the lowest MICs values and lowest median MIC. For the reference strain *S. aureus* ATCC 29213, the MIC of artificial honey was relatively high, indicating that osmotic activity is unlikely to be a dominant antimicrobial factor at concentrations at, or below 30% honey. The moderate correlation between MIC and MGO content, and the relatively high MIC of MGO alone, suggest that MGO may have only a modest impact on *S. aureus*. The lack of correlation between *S. aureus* MICs and any of the other physicochemical factors quantified in this study may indicate that characteristics or components other than those quantified here, may in fact be driving the antibacterial activity against this particular *S. aureus* strain. In contrast to *S. aureus*, the remaining Gram positive test strain *E. faecalis* ATCC 29212 was the most tolerant of the four test strains to manuka honeys, with many off-scale MICs (>30% w/v). Despite these off-scale results, a strong correlation was found between *E. faecalis* MICs and MGO content. Although this *E. faecalis* strain was even less susceptible to MGO alone than *S. aureus*, and was also not inhibited by artificial honey at a concentration of 30%, the strong correlation suggests that MGO may be an important driver of the activity of manuka honeys against this *E. faecalis* strain. Further testing with additional Gram positive species and strains is required to support these hypotheses.

For the Gram negative test organisms, MICs for *E. coli* ATCC 25922 correlated strongly with MGO content, suggesting that for this strain, MGO is a dominant antibacterial factor within manuka honeys. Compared to the Gram positive organisms, the susceptibility of *E. coli* to MGO was similar, however, *E. coli* was more susceptible to the osmotic activity of honey, with a reduction on OD of >90% at 30% artificial honey. The susceptibility of *P. aeruginosa* ATCC 27853 was similar to the *E. coli* strain in terms of osmotic activity, but it was the least susceptible to MGO, and MICs of manuka honey showed no correlation with MGO content. Previous studies have shown Gram negative bacteria to be more susceptible to the osmotic effects of honeys than Gram positive bacteria [5], with MICs for relatively low activity honeys, or artificial honey, generally not exceeding 30% honey [5,9,40]. Gram negative bacteria may also be less susceptible to small antibacterial molecules within honeys compared to the *S. aureus* strain, as in the current study, MICs for the Gram negative bacteria were always higher than those for *S. aureus*. The difference in susceptibility between the *E. coli* and *P. aeruginosa* strains may be due to the comparatively low permeability of the outer membrane of the *P. aeruginosa* species [41], or the capacity for *P. aeruginosa* to detoxify MGO [42]. Whilst *P. aeruginosa* strains are well-known to use efflux pumps as a tolerance strategy for many different antimicrobial agents, data show that MGO is in fact not a substrate that is recognised by *P. aeruginosa* efflux pumps [43]. Whilst the mechanisms of antibacterial action of MGO are not well studied, it is known to be a reactive dicarbonyl compound that interacts readily with proteins, and the cross-linking of proteins by MGO is thought to be a critical mechanism [6].

Examination of the relationship between MGO content and antibacterial activity using checkerboard assays showed that interactions were additive in nature, and that synergy did not occur for the organisms tested. Our experiments with MGO-amended honey found a similar trend, whereby the addition of increasing concentrations of MGO to multifloral honey resulted in corresponding, stepwise increases in antibacterial activity. Previous studies have also demonstrated additive antibacterial activity after the addition of MGO to honey [2,9,44]. MGO has been shown to have synergistic activity with other antimicrobial compounds such as linezolid against *S. aureus* [45], chitosan against *E. coli* and *P. aeruginosa* [46] and piperacillin,

amikacin and carbenicillin against *P. aeruginosa* [47]. To the best of our knowledge, no studies have previously investigated synergy between MGO and the honey matrix.

In summary, many studies have investigated the antimicrobial activity of manuka honey, however, this is the first to investigate the activity using a number of test methods, across a range of both honey samples and bacterial species. The data indicate that MGO content influences antibacterial activity, and that interactions appear to be largely additive in nature. It remains to be determined whether higher antibacterial activity in vitro translates into better clinical outcomes in a clinical, therapeutic context.

## Supporting information

**S1 Table.**
(DOCX)

## Acknowledgments

We acknowledge that this work was conducted on Noongar land, and that Noongar people remain the spiritual and cultural custodians of their land, and continue to practice their values, languages, beliefs and knowledge. We pay our respects to the traditional owners of the lands on which we live and work across Western Australia and Australia.

## Author Contributions

**Conceptualization:** Kathryn J. Green, Katherine A. Hammer.

**Data curation:** Kathryn J. Green, Katherine A. Hammer.

**Formal analysis:** Kathryn J. Green, Katherine A. Hammer.

**Funding acquisition:** Katherine A. Hammer.

**Investigation:** Kathryn J. Green, Ivan L. Lawag.

**Project administration:** Cornelia Locher, Katherine A. Hammer.

**Resources:** Cornelia Locher, Katherine A. Hammer.

**Supervision:** Cornelia Locher, Katherine A. Hammer.

**Visualization:** Kathryn J. Green.

**Writing – original draft:** Kathryn J. Green, Katherine A. Hammer.

**Writing – review & editing:** Kathryn J. Green, Cornelia Locher, Katherine A. Hammer.

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
