## [Decision Letter · Decision Letter 0]

7 Jul 2022

PONE-D-22-16741Correlation of the antibacterial activity of commercial manuka and Leptospermum honeys from Australia and New Zealand with methylglyoxal content and other physicochemical characteristicsPLOS ONE

Dear Dr. Hammer,

Thank you for submitting your manuscript to PLOS ONE. After careful consideration, we feel that it has merit but does not fully meet PLOS ONE’s publication criteria as it currently stands. Therefore, we invite you to submit a revised version of the manuscript that addresses the points raised during the review process.

We look forward to receiving your revised manuscript.

Kind regards,

Abdelwahab Omri, Pharm B, Ph.D, Laurentian University, Canada

Academic Editor

PLOS ONE

Journal Requirements:

Reviewers' comments:

Reviewer's Responses to Questions

**Comments to the Author**

1. Is the manuscript technically sound, and do the data support the conclusions?

Reviewer #1: Yes

Reviewer #2: Yes

Reviewer #3: Partly

Reviewer #4: Yes

2. Has the statistical analysis been performed appropriately and rigorously? 

Reviewer #1: Yes

Reviewer #2: N/A

Reviewer #3: No

Reviewer #4: Yes

3. Have the authors made all data underlying the findings in their manuscript fully available?

Reviewer #1: Yes

Reviewer #2: Yes

Reviewer #3: Yes

Reviewer #4: Yes

4. Is the manuscript presented in an intelligible fashion and written in standard English?

Reviewer #1: Yes

Reviewer #2: Yes

Reviewer #3: Yes

Reviewer #4: Yes

5. Review Comments to the Author

Reviewer #1: The present work is very interesting as it characterises 29 manuka/Leptospermum honeys and tries to associate to their content the potential antibiotical properties usually ascribed to them. The paper is well-structured and the English form is good. I have just a few comments to the authors for improving their manuscript.

a) The patronymic of the scientific names of the plant species mentioned in the text should be reported.

b) In the introduction or better in the discussion section, the authors should report the recent discoverey that honey also contain microRNA from gathered plants and bees and that also these coumpounds can also exert a potential biological role (antibiotical for instance?). This innovative aspect should be mentioned and commented with its own future perspectives. See and cite: BMC genomics, 2021, 22.1: 1-14; PLoS One, 2017, 12.2: e0172981; Food Chemistry: Molecular Sciences, 2021, 2: 100014.

c) How did the authors select the bacterial species to be tested?

d) In table 1, the unit of measure for the antibacterial activity value should be indicated

e) In general, the captions of the tables should be more descriptive of the content of the relative table.

f) asterisks indicating the significance of the data, compared to the control, should be reported in Fig 3

g) can be figure 1 and 2 made in color and not in scale of grey?

Reviewer #2: This is an interesting paper about manuka honeys. In fact, it is a paper on a highly competitive field, however, presents some quite surprising findings. Therefore it is publishable. In order to beneficiary improve the paper some changes in its layout are acknowledged. They are as follows (in random order);

1./ Table 1 characterizes the studied honeys. since the differences in MGO content provided by producers are unormously high in some samples (the same considers this content determined by Authors and presented in Table 2) there would be desirable to provide melissopalynologal data, at least for honeys showing extreme values;

2./ multifloral honey should be more detaily characterized in Experimental (geographic origin, producer ect.) while for artificial one producer should be identified;

3./ In Table 5 MIC data should be rather given not % data. If Author prefer the latter the detailed meaning should be given - for example 9% means high antibacterial activity but doean not give the reader the idea how strong is it.

Other small comments are as follows (in order of appearance):

a./ line 75 - please provide proper citation;

b./ Table 1 - is there really level of MGO content in MN02 sample was given as "0" by the producer? Or is it not given (on the other hand this honey seems as not being manuka also from data received by Authors;

c./ line 108 - were not was;

d./ line 130 - what Author mean by "aliquatos";

e./ line 205 -it is better to write that multifloral and artificial honeys serve as some kind of controls;

f./ line 245 E. coli should be in italics;

g./ line 249 -what Authors mean by statistical comparisons. In my opinion statistics was not performed at all;

h./ line 266 - do Authors mean that generation of hydrogen peroxide was negligible or small in comparison with other honeys?

i./ lines 403-403 - HMF level in regions of higher temperature is often quite high;

m./ line 440: do Authors mean that osmotic activity is of lower meaning that MGO content and thus manuka honeys antimicrobial acrion comes from appropriate lebvel of MGO?

n./ it would be beneficial id heat maps would be coloured.

Reviewer #3: - Abstract need more revision to be attractive for the reader.

- References should be updated.

- statistical analysis must be done for all your manuscript results.

- Figures are not clear you should be represented in higher resolution.

Reviewer #4: This study reviews the characteristics of 30 honeys and their antimicrobial activity against 4 type strains of bacteria. It is a comprehensive analysis of many honey samples that convincingly demonstrates the variability between the many manuka and Leptospermum honeys on the market. Specific comments are as follows:

-Despite testing four species of organism, only 1 strain of each was tested and four is far from a comprehensive range of bacterial pathogens, and is not that far different from testing one strain. Please tone down the language in lines 24, 485-486. Also in the Discussion, the authors should not overgeneralize their findings related to organism species, as only 1 strain of each of 4 species was tested. Please revise the language in lines 443-473.

-Honeys were analysed within three months of acquisition (page 5). It would also be informative to include the duration of time that elapsed between the quantitation of physicochemical parameters and measurements of antimicrobial activity.

-Please provide a clear rationale for testing synergy of MGO with honey, as MGO is already known to be the main antimicrobial component of manuka honey, and the relationship between its concentration and antimicrobial activity has already been established. Synergy (e.g., line 482) seems not to be the appropriate term to use. Lines 341-343 data should be presented perhaps as a supplemental Table.

6. PLOS authors have the option to publish the peer review history of their article (what does this mean?). If published, this will include your full peer review and any attached files.

Reviewer #1: No

Reviewer #2: **Yes: **Paweł Kafarski

Reviewer #3: No

Reviewer #4: No

---

## [Author Response · Author response to Decision Letter 0]

16 Jul 2022

PONE-D-22-16741

Correlation of the antibacterial activity of commercial manuka and Leptospermum honeys from Australia and New Zealand with methylglyoxal content and other physicochemical characteristics

Response to Reviewers' comments:

Reviewer #1: 

The present work is very interesting as it characterises 29 manuka/Leptospermum honeys and tries to associate to their content the potential antibiotical properties usually ascribed to them. The paper is well-structured and the English form is good. I have just a few comments to the authors for improving their manuscript.

a) The patronymic of the scientific names of the plant species mentioned in the text should be reported.

Response: We have added this information to the manuscript. 

b) In the introduction or better in the discussion section, the authors should report the recent discoverey that honey also contain microRNA from gathered plants and bees and that also these coumpounds can also exert a potential biological role (antibiotical for instance?). This innovative aspect should be mentioned and commented with its own future perspectives. See and cite: BMC genomics, 2021, 22.1: 1-14; PLoS One, 2017, 12.2: e0172981; Food Chemistry: Molecular Sciences, 2021, 2: 100014.

Response: Thank you for alerting us to this discovery. This information has been added to the discussion. 

c) How did the authors select the bacterial species to be tested?

Response: These organisms are the quality control reference strains recommended by the Clinical and Laboratory Standards Institute and the European Committee on Antimicrobial Susceptibility Testing. This information has been added to the text.

d) In table 1, the unit of measure for the antibacterial activity value should be indicated

Response: The AAV does not have units, much like pH and refractive index do not have units. 

e) In general, the captions of the tables should be more descriptive of the content of the relative table.

Response: The table titles have been updated to be more informative. 

f) asterisks indicating the significance of the data, compared to the control, should be reported in Fig 3

Response: Given that all data points differ significantly from the control at each time point (for both organisms), we have elected not to add asterisks to the figures. Instead, we have added this information to the figure caption. 

g) can be figure 1 and 2 made in color and not in scale of grey?

Response: We agree that colour figures would be far more visually appealing than grey scale. That said, the black/white scale offers the largest range and intensity of shading, is most uniformly represented on computer LCD screens, can be printed most easily, and is not subject to misinterpretation due to an individual’s visual circumstances, such as colour blindness. 

Reviewer #2: 

This is an interesting paper about manuka honeys. In fact, it is a paper on a highly competitive field, however, presents some quite surprising findings. Therefore it is publishable. In order to beneficiary improve the paper some changes in its layout are acknowledged. They are as follows (in random order);

1./ Table 1 characterizes the studied honeys. since the differences in MGO content provided by producers are unormously high in some samples (the same considers this content determined by Authors and presented in Table 2) there would be desirable to provide melissopalynologal data, at least for honeys showing extreme values;

Response: Commercial manuka honeys are well known to contain a broad range of MGO content and the collection examined in this paper represents this range. The authors agree that pollen analysis may provide additional insight into the honeys examined, but unfortunately pollen analysis is beyond the expertise of our research team and potentially beyond the scope of this project, which was focussed on correlating antibacterial activity with MGO content. 

2./ multifloral honey should be more detaily characterized in Experimental (geographic origin, producer ect.) while for artificial one producer should be identified;

Response: More details have been added for multifloral honey. The artificial honey is prepared in the laboratory as described in the materials section, and as such does not have a producer. 

3./ In Table 5 MIC data should be rather given not % data. If Author prefer the latter the detailed meaning should be given - for example 9% means high antibacterial activity but doean not give the reader the idea how strong is it.

Response: The values shown in the table are already MICs. The MICs are expressed as percentage of honey (i.e. the unit of measurement is percentage). This has been clarified in the table footnote. These data are interpreted by comparing the MICs of MGO or honey alone, to MGO in combination with honey. 

Other small comments are as follows (in order of appearance):

a./ line 75 - please provide proper citation;

Response: We have checked the in-text citation and believe that it is already correct. 

b./ Table 1 - is there really level of MGO content in MN02 sample was given as "0" by the producer? Or is it not given (on the other hand this honey seems as not being manuka also from data received by Authors;

Response: Thank you for bringing this to our attention. We have corrected this to “not stated”.

c./ line 108 - were not was;

Response: It is grammatically correct to use “was” in this sentence, as in “...the content was determined...”.

d./ line 130 - what Author mean by "aliquatos";

Response: Aliquot is another way of saying “sample”. Since this term is commonly used in scientific writing we have elected to leave it in the manuscript. 

e./ line 205 -it is better to write that multifloral and artificial honeys serve as some kind of controls;

Response: Thank you; we have added to the text that these honeys served as comparators. 

f./ line 245 E. coli should be in italics;

Response: Thank you; this has been italicised. 

g./ line 249 -what Authors mean by statistical comparisons. In my opinion statistics was not performed at all;

Response: This wording of the statistics section has been clarified to make it clearer that whilst we did many statistical comparisons, we could not compare honeys from different countries, or from different floral sources because there were not enough samples in each group. 

h./ line 266 - do Authors mean that generation of hydrogen peroxide was negligible or small in comparison with other honeys?

Response: The sentence describing hydrogen peroxide content has been rephrased to clarify this. 

i./ lines 403-403 - HMF level in regions of higher temperature is often quite high;

Response: New Zealand and the regions of Australia where manuka may be harvested are temperate climates and as such high HMF levels would not be expected. We have added some text stating the allowable HMF limits in tropical climates. 

m./ line 440: do Authors mean that osmotic activity is of lower meaning that MGO content and thus manuka honeys antimicrobial acrion comes from appropriate lebvel of MGO?

Response: Yes, there is a clear relationship between MGO content and antibacterial activity although this varies between bacterial species and the particular antibacterial test used. 

n./ it would be beneficial id heat maps would be coloured.

Response: We agree that colour figures would be far more visually appealing than grey scale. That said, the black/white scale offers the largest range and intensity of shading, is most uniformly represented on computer LCD screens, is easily printed and is not subject to misinterpretation due to an individual’s visual circumstances, such as colour blindness. 

Reviewer #3: 

- Abstract need more revision to be attractive for the reader.

Response: The abstract has been revised to make it more appealing. 

- References should be updated.

Response: The most relevant recent references have already been included in the manuscript. If the reviewer believes that specific references have been omitted we would be happy to include these. 

- statistical analysis must be done for all your manuscript results.

Response: Statistical analyses have been performed where appropriate. The section describing the statistical analyses has been updated to reflect this. 

- Figures are not clear you should be represented in higher resolution.

Response: Thank you: high resolution figures have been submitted. 

Reviewer #4: 

This study reviews the characteristics of 30 honeys and their antimicrobial activity against 4 type strains of bacteria. It is a comprehensive analysis of many honey samples that convincingly demonstrates the variability between the many manuka and Leptospermum honeys on the market. Specific comments are as follows:

-Despite testing four species of organism, only 1 strain of each was tested and four is far from a comprehensive range of bacterial pathogens, and is not that far different from testing one strain. Please tone down the language in lines 24, 485-486. 

Response: Thank you: the text has been modified to place less emphasis on the number of test strains. 

Also in the Discussion, the authors should not overgeneralize their findings related to organism species, as only 1 strain of each of 4 species was tested. Please revise the language in lines 443-473.

Response: The text has been modified to clarify that discussions relate to the strains tested in the current study, and not to all strains of each entire species. 

-Honeys were analysed within three months of acquisition (page 5). It would also be informative to include the duration of time that elapsed between the quantitation of physicochemical parameters and measurements of antimicrobial activity.

Response: It would be very time consuming to track back through all of the testing records to determine the numbers of days or weeks elapsed between each of the tests. Honeys are generally very stable when stored at room temperature for periods of three months or less so the authors are confident that minimal changes would have occurred during the 3 month testing window. In addition, many of these honeys may have been sitting on the shelves of shops or warehouses for months or years before testing so would have already been relatively mature honeys. 

-Please provide a clear rationale for testing synergy of MGO with honey, as MGO is already known to be the main antimicrobial component of manuka honey, and the relationship between its concentration and antimicrobial activity has already been established. 

Response: Interestingly, the relationship between MGO and manuka antibacterial activity has only been established for one reference strain of Staphylococcus aureus, using an agar diffusion assay. The relationship for other organisms, or determined using non-agar methods, has not been investigated which was part of the reason for this study. 

Inspection of MIC data for manuka honeys showed that the actual amount of MGO that would have been present at each MIC was typically about one quarter of the MIC of MGO alone, and the MIC also occurred at concentrations well below those where osmotic activity would be having a direct antibacterial effect. We therefore wondered if a synergistic interaction was occurring between the honey matrix and the MGO. We apologise for omitting this rationale in the manuscript and have added words to this effect to the methods section. 

Synergy (e.g., line 482) seems not to be the appropriate term to use. 

Response: We have inspected these cited publications carefully and in our opinion the term ‘synergy’ has been used appropriately and is well defined. 

Lines 341-343 data should be presented perhaps as a supplemental Table.

Response: This data is already presented in Table 5.  

Comments to the Author

1. Is the manuscript technically sound, and do the data support the conclusions?

Reviewer #1: Yes

Reviewer #2: Yes

Reviewer #3: Partly

Reviewer #4: Yes

2. Has the statistical analysis been performed appropriately and rigorously? 

Reviewer #1: Yes

Reviewer #2: N/A

Reviewer #3: No

Reviewer #4: Yes

3. Have the authors made all data underlying the findings in their manuscript fully available?

Reviewer #1: Yes

Reviewer #2: Yes

Reviewer #3: Yes

Reviewer #4: Yes

4. Is the manuscript presented in an intelligible fashion and written in standard English?

Reviewer #1: Yes

Reviewer #2: Yes

Reviewer #3: Yes

Reviewer #4: Yes

6. PLOS authors have the option to publish the peer review history of their article (what does this mean?). If published, this will include your full peer review and any attached files.

Do you want your identity to be public for this peer review? For information about this choice, including consent withdrawal, please see our Privacy Policy.

Reviewer #1: No

Reviewer #2: Yes: Paweł Kafarski

Reviewer #3: No

Reviewer #4: No

---

## [Editor Report · Decision Letter 1]

19 Jul 2022

Correlation of the antibacterial activity of commercial manuka and Leptospermum honeys from Australia and New Zealand with methylglyoxal content and other physicochemical characteristics

PONE-D-22-16741R1

Dear Dr. Katherine Ann Hammer,

We’re pleased to inform you that your manuscript has been judged scientifically suitable for publication and will be formally accepted for publication once it meets all outstanding technical requirements.

Kind regards,

Abdelwahab Omri, Pharm B, Ph.D, Laurentian University, Canada

Academic Editor

PLOS ONE

---

## [Editor Report · Acceptance letter]

21 Jul 2022

PONE-D-22-16741R1 

Correlation of the antibacterial activity of commercial manuka and *Leptospermum* honeys from Australia and New Zealand with methylglyoxal content and other physicochemical characteristics 

Dear Dr. Hammer:

I'm pleased to inform you that your manuscript has been deemed suitable for publication in PLOS ONE. Congratulations! Your manuscript is now with our production department. 

Kind regards, 

on behalf of

Dr. Abdelwahab Omri 

Academic Editor

PLOS ONE